# Brain-Specific Angiogenesis Inhibitor 3 Is Expressed in the Cochlea and Is Necessary for Hearing Function in Mice

**DOI:** 10.3390/ijms242317092

**Published:** 2023-12-04

**Authors:** Chika Saegusa, Wataru Kakegawa, Eriko Miura, Takahiro Aimi, Sachiyo Mogi, Tatsuhiko Harada, Taku Yamashita, Michisuke Yuzaki, Masato Fujioka

**Affiliations:** 1Department of Molecular Genetics, Kitasato University School of Medicine, Kanagawa 252-0374, Japan; saegusa.chika@kitasato-u.ac.jp; 2Department of Otorhinolaryngology, Head and Neck Surgery, Keio University School of Medicine, Tokyo 160-8582, Japan; 3Department of Physiology, Keio University School of Medicine, Tokyo 160-8582, Japan; wkake@z7.keio.jp (W.K.); erk-miura@keio.jp (E.M.); t.aimi.1116@keio.jp (T.A.); myuzaki@keio.jp (M.Y.); 4Department of Otorhinolaryngology, Head and Neck Surgery, Kitasato University, Kanagawa 252-0374, Japan; mogi@med.kitasato-u.ac.jp (S.M.); tyamahns@kitasato-u.ac.jp (T.Y.); 5Department of Otolaryngology, International University of Health and Welfare, Shizuoka 413-0012, Japan; t-harada@iuhw.ac.jp; 6Clinical and Translational Research Center, Keio University Hospital, Tokyo 162-8582, Japan

**Keywords:** cochlea, hair cell, hearing, knockout mice, pillar cell, spiral ganglion neuron

## Abstract

Mammalian auditory hair cells transduce sound-evoked traveling waves in the cochlea into nerve stimuli, which are essential for hearing function. Pillar cells located between the inner and outer hair cells are involved in the formation of the tunnel of Corti, which incorporates outer-hair-cell-driven fluid oscillation and basilar membrane movement, leading to the fine-tuned frequency-specific perception of sounds by the inner hair cells. However, the detailed molecular mechanism underlying the development and maintenance of pillar cells remains to be elucidated. In this study, we examined the expression and function of brain-specific angiogenesis inhibitor 3 (Bai3), an adhesion G-protein-coupled receptor, in the cochlea. We found that Bai3 was expressed in hair cells in neonatal mice and pillar cells in adult mice, and, interestingly, *Bai3* knockout mice revealed the abnormal formation of pillar cells, with the elevation of the hearing threshold in a frequency-dependent manner. Furthermore, old *Bai3* knockout mice showed the degeneration of hair cells and spiral ganglion neurons in the basal turn. The results suggest that Bai3 plays a crucial role in the development and/or maintenance of pillar cells, which, in turn, are necessary for normal hearing function. Our results may contribute to understanding the mechanisms of hearing loss in human patients.

## 1. Introduction

Hearing loss is one of the most common sensory impairments in humans [1]. Sensorineural hearing loss, which is caused by damage to the inner ear cells or auditory nerve, is difficult to treat because most damaged inner ear cells cannot be regenerated in adult humans. Understanding the molecular mechanisms that regulate the development and maintenance of the mammalian inner ear structure and/or function is important for identifying novel therapeutic targets for sensorineural hearing loss.

Auditory information is received and converted into electrical signals in cochlear hair cells (HCs) in the inner ear, which then activate spiral ganglion neurons (SGNs), and they are transmitted to the auditory center in the brain [2]. SGNs are bipolar type I and type II afferent neurons innervating inner hair cells (IHCs) and outer hair cells (OHCs), respectively. HCs and SGNs are essential for auditory function, and the impairment or loss of HCs and SGNs is responsible for hearing loss. In addition to HCs and SGNs, pillar cells (PCs), which are arranged between IHCs and OHCs, are important structural elements of the organ of Corti. PCs form the tunnel of Corti, which plays a critical role in hearing function. Impaired PCs result in a collapsed tunnel of Corti and hearing loss [3]. Recently, it was reported that the disorganization of microtubules in PCs causes reduced PC stiffness, the disruption of the cytoskeletal architecture of the cochlea, and consequent hearing loss in mice [4]. Despite the importance of PCs in hearing function, the molecular context remains to be elucidated.

Brain-specific angiogenesis inhibitor 3 (Bai3) is an adhesion G-protein-coupled receptor [5], and its extracellular domain binds to C1q-like (C1ql) proteins [6]. The interaction between C1ql1, as the ligand, and Bai3, as the receptor, plays an indispensable role in cerebellar synapse organization [7,8]. In the olfactory bulb, Bai3 transduces signals for synapse formation by binding to the ligand, the pre-synaptic C1ql3 protein [9]. Outside the central nervous system, Bai3 also plays diverse physiological roles. Bai3 mediates the inhibitory effects of C1ql3 on insulin secretion from pancreatic beta cells [10]. In myoblasts, Bai3 promotes the fusion process, which is negatively regulated by C1ql4 [11]. Conversely, the intracellular domain of Bai3 binds to ELMO1, which is implicated in cytoskeletal remodeling [12]. The interaction between Bai3 and ELMO regulates the fusion process in myoblasts [11,13] and dendrite morphogenesis in Purkinje cells [14].

Bai3 and C1ql1 have also been suggested to be expressed in the cochlea during inner ear development by systemic transcriptome analysis (SHIELD database “https://shield.hms.harvard.edu, doi:10.1093/database/bav071 (accessed on 21 October 2021)” [15]), and are expected to play a role in the auditory system. Recently, C1ql1 was reported to be expressed in the mouse cochlea. Biswas et al. determined that C1ql1 is expressed in a subset of OHCs in the adult mouse cochlea by using a fluorescent reporter gene inserted into the C1ql1 locus [16]. Another group showed that C1ql1 is expressed in hair cells and spiral ganglion neurons by immunostaining with an anti-C1ql1 antibody [17]. It is possible that Bai3 plays a role as a receptor for C1ql1 in the auditory system as well as in other tissues. However, the expression profiles and the role of Bai3 in the auditory system remain unclear. Analysis of the function of Bai3 in the mouse auditory system may help to understand the molecular mechanisms of the development and functions of the auditory system.

In this study, we analyzed the expression of Bai3 in the mouse cochlea and the hearing function of *Bai3* knockout mice to study the function of Bai3 in the auditory system.

## 2. Results

### 2.1. Bai3 Is Dispensable for Hair Cell Development in the Neonatal Cochlea

We examined the endogenous expression pattern of Bai3 in mouse cochleae via immunohistochemistry. Bai3 immunosignals were detected in hair cells within the embryonic and neonatal cochleae of *Bai3*^+/+^ mice (arrows in Figure 1A and Appendix A), suggesting that Bai3 is expressed in developing hair cells. To examine the effects of the absence of Bai3 on hair cell development, we performed whole-mount immunostaining of the neonatal cochlear tissue with an antibody to MYO7a, a marker for HCs, and phalloidin and compared the staining pattern of *Bai3*^−/−^ cochleae with that of *Bai3*^+/+^ cochleae. Despite the expression of Bai3 in HCs during development, the staining patterns of MYO7a and phalloidin were not significantly different between *Bai3^−/−^* and *Bai3*^+/+^ mice, and we did not observe any developmental abnormalities in the HCs of *Bai3^−/−^* mice (Figure 2A,B).

We also examined the expression of C1ql1 as a ligand candidate for Bai3 in the cochlea at P3 using *C1ql1^+/GFP^* mice [7]. In these mice, cells expressing C1ql1 also expressed green fluorescent protein (GFP). Immunosignals of GFP in *C1ql1^+/GFP^* cochlea at P3 were observed in neurons innervating the IHCs (Appendix A). Based on the expression of C1ql1 in neurons innervating the basal membrane of IHCs (Appendix A), which express Bai3, we hypothesized that the interaction between C1ql1 and Bai3 plays a role in the refinement of neuronal innervation in HCs. We examined the innervation patterns of neurons that were positive for neurofilament H (Figure 2C) and peripherin (Figure 2D) into HCs in the cochlea at P3. However, we did not find any abnormal innervation patterns of neurons in either the *Bai3^−/−^* or *C1ql1^−/−^* cochlea (Figure 2C,D; Appendix A), suggesting that neither Bai3 nor C1ql1 is indispensable in the innervation of neurons to HCs at this stage.

### 2.2. Bai3 Is Expressed in Adult Pillar Cells in the Mouse Cochlea

The expression of Bai3 in hair cells is decreased in adult mice. Contrastingly, in adult mice, Bai3 immunosignals were detected in PCs (Figure 1B), but not in adult *Bai3^−/−^* pillar cells (Appendix A).

### 2.3. High-Frequency Dominant Hearing Loss in Bai3^−/−^ Mice

To examine the function of Bai3 in the auditory system, we compared the ABRs in *Bai3*^+/+^ and *Bai3^−/−^* mice (Figure 3A–C). *Bai3^−/−^* mice showed increased thresholds at 16, 24, and 32 kHz compared to *Bai3*^+/+^ mice (Figure 3A), although the threshold was not significantly different between *Bai3*^+/+^ and *Bai3^−/−^* mice at 8 kHz. Representative results from the ABR waves at 24 kHz in 14-week-old mice are shown in Figure 3B. Wave I represents neuronal activation in spiral neurons in the cochlea, and wave II and beyond represent neuronal activation in the central nervous system (CNS). *Bai3^−/−^* mice showed altered waveforms of waves I–III (Figure 3B), and the average wave I amplitude was significantly lower in *Bai3^−/−^* mice than in *Bai3*^+/+^ mice (*p* < 0.001, Figure 3C). These results suggest that the impaired ABR observed in *Bai3^−/−^* mice is generated, at least in part, in the cochlea and/or spiral ganglion neurons. We found that some old *Bai3^−/−^* mice at 17 months of age (seven ears from four mice) showed no measurable response at the highest stimulus levels tested. To investigate the deficits in OHC electromotility, we assessed the DPOAEs. DPOAE testing revealed increased thresholds at 16 and 24 kHz in *Bai3^−/−^* mice compared to *Bai3*^+/+^ mice (Figure 3D,E), although the threshold was not significantly different at 8 kHz between *Bai3*^+/+^ and *Bai3^−/−^* mice. We also examined the hearing ability of young (14–17 weeks of age) and old adult mice (17 months of age). Regardless of age, *Bai3^−/−^* mice showed elevated thresholds at 16 kHz and higher frequencies compared to *Bai3*^+/+^ mice in both the ABR and DPOAE. These results suggest that *Bai3^−/−^* mice have hearing impairment at high frequencies.

### 2.4. Pillar Cells Are Thinner in Bai3^−/−^ Mice

To investigate the mechanisms underlying hearing impairment in *Bai3^−/−^* mice, we performed a histological analysis of adult *Bai3^−/−^* mice and found that the stalks of the outer pillar cells were thinner in *Bai3^−/−^* mice compared to those of *Bai3*^+/+^ mice, suggesting that the pillar cells were not properly formed in the cochleae of *Bai3^−/−^* mice (Figure 4A,B). These results suggest that Bai3 expression is required for the development and/or maintenance of pillar cells. As the structural organization of cochlear cells is supported by cytoskeleton proteins, we examined the actin cytoskeleton by immunostaining, but we did not find any abnormal staining of beta-actin or phalloidin in *Bai3^−/−^* mice (Figure 4C and Appendix A) compared to *Bai3*^+/+^ mice. Furthermore, we did not find any significant differences in the expression pattern of acetylated tubulin, which PCs are composed of, in *Bai3^−/−^* PCs compared to *Bai3*^+/+^ PCs (Figure 4C).

### 2.5. Hair Cells and Spiral Ganglion Neurons Are Degenerated in the Basal Turn of Old Bai3^−/−^ Mice

While examining the hearing ability of *Bai3^−/−^* mice, we found that some old *Bai3^−/−^* mice showed no measurable ABR. We fixed such ‘scale out’ *Bai3^−/−^* mice, performed a histological analysis, and compared the results with those of age-matched *Bai3*^+/+^ mice. We observed a loss of MYO7a-positive HCs in the basal turn of *Bai3^−/−^* mice but not in age-matched *Bai3*^+/+^ mice (right in Figure 5A,B). In the middle turn, MYO7a-positive IHCs were observed in *Bai3^−/−^* mice as well as in *Bai3*^+/+^ mice (left in Figure 5A,B); however, OHCs were not clearly identified, probably due to the abnormal structure of the organ of Corti in *Bai3^−/−^* mice. We also found significant degeneration of TUBB-positive cells in Rosenthal’s canal at the basal turn in *Bai3^−/−^* mice (*p* < 0.0001, Figure 5C,D). We did not detect any specific immunosignals of Bai3 in SGNs in any of the mice we examined, suggesting that the degeneration of SGNs in *Bai3^−/−^* mice is not the primary phenotype caused by the deletion of the *Bai3* gene.

## 3. Discussion

We found that Bai3 is expressed in the mouse inner ear, and that the hearing threshold is increased at high frequencies in *Bai3^−/−^* mice. Pillar cells in *Bai3^−/−^* mice were thinner than those in *Bai3*^+/+^ mice. The degeneration of hair cells and spiral ganglion neurons was also observed in old *Bai3^−/−^* mice. These results suggest that Bai3 is required for normal hearing function in mice.

HCs at the basal turn detect high-frequency sounds, whereas those at the apical turn detect low-frequency sounds [18]. In old *Bai3^−/−^* mice, we observed the loss of HCs and the degeneration of SGNs in the basal to midbasal turn, but not in the middle and apical turns. These results indicate that the frequency-dependent hearing impairment and region-specific cell degeneration observed in *Bai3^−/−^* mice were consistent. However, the elevation of the ABR thresholds at high frequencies was also apparent in young adult *Bai3^−/−^* mice at 4 and 17 weeks of age, although histological abnormalities such as the degeneration of HCs and SGNs were not observed in young adult *Bai3^−/−^* mice. These results suggest that the degeneration of HCs and SGNs is not a primary cause of the elevated hearing threshold in *Bai3^−/−^* mice. We also have to consider that the mice used in this study were on a C57BL6 background and thus carry a mutation in the *cdh23* gene, and hearing loss appears at a younger age than in other strains [19]. It is possible that the phenotypes reported in this study are affected by a mutation in the *cdh23* gene. Future studies using *Bai3^−/−^* mice on other genetic backgrounds, such as CBA or C57BL6, in which the mutation site in the *cdh23* gene was corrected by genome editing [19], may help us to understand the roles of *Bai3* in auditory functions.

To understand the primary effect of Bai3 deficiency in the cochlea, we focused on the cytoskeleton in *Bai3^−/−^* cochleae. Previously, it was reported that the intercellular domain of Bai3 interacts with ELMO [13,14]. Bai3 and ELMO also bind to the dedicator of cytokinesis 1 (Dock180) to activate the Rac pathway in neurons and are involved in the reconstitution of the cytoskeleton in Purkinje cells [14]. These results suggest that Bai3 is involved in cytoskeletal organization. Cochlear pillar cells, in which Bai3 is expressed, harbor a distinctive cytoskeletal architecture with long bundles of microtubules integrated with actin filaments [20,21]. The number of microtubules is closely correlated with the stiffness of PCs, which are responsible for the proper transmission of auditory signals. These results led us to hypothesize that Bai3 regulates cytoskeletal organization in PCs. However, we did not find any abnormal staining of acetylated tubulin or actin in *Bai3^−/−^* PCs compared to that in *Bai3*^+/+^ under our experimental conditions.

Although we did not find any altered immunostaining for the cytoskeleton in *Bai3^−/−^* mice in this study, it is still possible that Bai3 is implicated in the structure of the organ of Corti, because we found not only thinner pillar cells but also collapsed tunnel of Corti in some *Bai3^−/−^* mice (two out of five) at the age of 9–11 weeks (Appendix A). Although individual phenotypic variation in the tunnel of Corti is observed, abnormal formation of the tunnel of Corti is not inconsistent with the phenotypes of thinner pillar cells in *Bai3^−/−^* mice. The tunnel of Corti has been implicated in OHC-driven fluid flow oscillation, which is required for hearing function [22] and structural maturation of the organ of Corti, including the opening of the tunnel of Corti during development. OHC-driven fluid flow oscillation is accompanied by the onset of hearing function [23], indicating that the proper structure of the cochlea, including the tunnel of Corti, is indispensable for transmitting mechanical sound stimuli. An abnormal tunnel of Corti and hearing impairment have been previously reported in several mutant mice. For example, *Fgfr3*-mutant mice showed a failure of PC differentiation and tunnel of Corti formation [3]. Moreover, the stiffness of *Fgfr3*-deficient PCs was only 50% of that of the control PCs [24]. *Fgfr3*-mutant mice also showed reduced innervation of OHCs and profound hearing loss. A dominant-negative *Gjb2* mutation in mice also resulted in a reduction in the number of microtubules in PCs and the failure of the opening of the tunnel of Corti during early postnatal stages such as P8 [25]. It has been suggested that Gjb2 is indispensable in the maturation of the tunnel of Corti. Notably, *Gjb2*-dominant-negative transgenic mice did not show ABRs. These results indicate the importance of the structure of the tunnel of Corti ensured by PCs in hearing function. Recently, it was reported that GAS2 protein harboring microtubule- and actin-binding domains was required to organize and stabilize the microtubules in PCs [4], and that *GAS2*-mutant mice showed a decrease in PC stiffness and hearing loss. Interestingly, both human patients with *GAS2* gene mutations and *GAS2*-mutant mice showed hearing loss, which was more pronounced at higher frequencies. The mechanisms underlying high-frequency dominant hearing impairment reported in *GAS2*-mutant mice are unknown. In this study, an elevated threshold was observed at high frequencies in *Bai3^−/−^* mice, although thinner pillar cells and an abnormal structure of the tunnel of Corti were observed from the basal region to the apex along the cochlear turn (Figure 4 and Appendix A). Even though the average thresholds of ABR were not altered at 8 kHz in *Bai3^−/−^* mice compared to *Bai3*^+/+^ mice, our DPOAE results showed slightly reduced activity even at 8 kHz in young adult *Bai3^−/−^* mice (Figure 3D). These results suggest that the lack of Bai3 protein affects hearing function at low and high frequencies, but the effect is more pronounced at higher frequencies. Since mechanical and electrical properties differ between the basal and apical regions of the cochlea, we have discussed the possible mechanisms underlying high-frequency dominant hearing impairment in *Bai3^−/−^* mice. (1) Abnormal formation of the pillar cells and the tunnel of Corti caused by *Bai3* deletion affects mechanics more severely in the basal region than in the apical region of the cochlea. PCs at the cochlear basal turn corresponding to higher frequencies have a greater number of microtubules than those at the apical turn corresponding to lower frequencies [26]; thus, the mechanical properties of the tunnel of Corti, and thereby the organ of Corti, are different depending on the region along the cochlear length [27]. It is suggested that the altered cytoskeletal structure of PCs affects the mechanics required to process sound stimulation differently depending on the region along the cochlear length, which may result in frequency-dependent hearing impairment. (2) The mechanisms by which basilar membrane vibrations are translated into neural excitation differ between the apical and basal regions of the cochlea [28]. It is possible that the synchronized vibration of the basilar membrane in a broad area compensates for the subtle impairment of the structure of the organ of Corti in the apex; however, in the basal region, subtle impairment of the structure of the organ of Corti affects sound processing more severely, resulting in an elevated hearing test threshold. Although the molecular roles of Bai3 in the cytoskeletal architecture were not elucidated in this study, further studies, such as electron microscopic analysis, may reveal the roles of Bai3 in the cytoskeletal architecture in the cochlea and the mechanisms of the frequency-dependent hearing impairment found in *Bai3^−/−^* mice.

Biswas et al. demonstrated that C1ql1 is expressed in a subset of OHCs in adult mice and also suggested that Bai3 is localized at the afferent post-synaptic sites of OHCs [16]. However, the Bai3 immunosignals at the afferent post-synaptic sites were below the detection limit under our experimental conditions. It is still possible that low levels of Bai3 play a role in synapse formation or maintenance by binding to C1ql1 in the cochlea of adult mice and may contribute to the hearing impairment in *Bai3^−/−^* mice. Alternatively, the abnormally formed pillar cells and tunnel of Corti could indirectly affect the formation or maintenance of synapses between OHCs and efferent nerves in *Bai3^−/−^* mice.

Mutations in the *Bai3* gene have been described in association with human diseases. The *Bai3* gene is implicated in psychiatric disorders such as schizophrenia and bipolar disorder [29,30,31,32,33,34], suggesting a role for Bai3 in the CNS. Bai3 was reported to be expressed in the cerebellum [7] and granule cells of the olfactory bulb [9] in the CNS, but its expression in the auditory pathway in the CNS remains unknown. It would be interesting to investigate the precise expression of Bai3 in the auditory pathway in the CNS, as well as in the peripheral organs and cochlea, to understand the role of Bai3 in auditory function. However, in our current ABR results, we did not find any changes in the latency between waves I and III or III–V, suggesting that Bai3 is dispensable in the auditory pathway in the CNS. Although a relationship between *Bai3* gene mutations and human patients with hearing loss has not been reported so far, our finding that Bai3 in the cochlea is required for the normal development of pillar cells, which are indispensable for hearing function, may help us to understand the pathogenesis of hearing loss and find novel therapeutic targets for hearing restoration in the future.

In conclusion, we showed that *Bai3* deficiency leads to the abnormal formation of pillar cells and an elevated hearing threshold at high frequencies. Furthermore, we showed the degeneration of HCs and SGNs in the basal turn of old *Bai3^−/−^* mouse cochleae. These novel findings suggest that Bai3 in the cochlea is required for normal hearing function.

## 4. Materials and Methods

### 4.1. Mice

*Bai3^−/−^* mice were generated as reported previously [7]. Briefly, exons 8 and 9 of the *Bai3* gene were deleted, resulting in a functional deficiency of the Bai3 protein. *Bai3^−/−^* mice were maintained on a C57BL/6 background.

### 4.2. Immunofluorescence

Adult anesthetized mice were transcardially perfused with phosphate-buffered saline (pH 7.4) and 4% PFA (paraformaldehyde; Nacalai Tesque, Inc., Kyoto, Japan). Cochleae were removed from the mice, perfused with 4% PFA through the round window, oval window, and apex, and then fixed for an additional 24 h in PFA at 4 °C. They were then decalcified for two days in decalcifying solution B (FUJI FILM, Osaka, Japan) at 4 °C. The cochleae from neonatal mice were dissected without transcardial perfusion and fixed with 4% PFA overnight at 4 °C. On embryonic day 18.5, the cochleae were fixed with 4% PFA for 2 h at 4 °C. These tissues were used for whole-mount or section immunostaining. For immunostaining using frozen sections, the cochleae were incubated in 10% sucrose for 1 h at 4 °C and then in 30% sucrose overnight at 4 °C. The cochleae were frozen using Tissue-Tek OCT compound (Sakura Finetek Japan Co., Ltd., Tokyo, Japan) and cut into 7 μm frozen sections. For immunostaining using paraffin sections, the cochleae were decalcified for 7 days in decalcifying solution B at 4 °C, and 4 μm paraffin sections were used for analysis. Antigen retrieval was performed using 10 mM citrate buffer (pH 6), and tissues were permeabilized with 0.3% Triton-X100 in phosphate-buffered saline. Tissues were blocked using normal donkey serum (Abcam, Cambridge, UK). Details of the antigen retrieval, permeabilization, and blocking methods are summarized in Appendix A. After incubation with primary antibodies overnight at 4 °C, the sections were incubated with the appropriate secondary antibodies conjugated with Alexa Fluor (1:500, Thermo Fisher Scientific, Waltham, MA, USA) for 1–2 h at room temperature. The following primary antibodies were used: guinea pig anti-Bai3 (1:100, A323 created in Kakegawa et al., 2015 [7]), mouse anti-MYO7a (1:30; DSHB, Iowa City, IA, USA), rabbit anti-MYO7a (1:100; Proteus Biosciences, Ramona, CA, USA), rabbit anti-SOX2 (1:200; Abcam), goat anti-SOX2 (1:200; R&D Systems, Minneapolis, MN, USA), chick anti-neurofilament-H (1:1000; Abcam), rabbit anti-peripherin (1:100; sigma, Burbank, CA, USA), mouse anti-acetylated tubulin (1:1000; sigma, Burbank, CA, USA), rabbit anti-beta-Actin (1:500; Cell Signaling Technology, Danvers, MA, USA) and mouse anti-TUBB3 (1:1000; Promega, Madison, WI, USA). Actin filaments were stained with phalloidin conjugated with rhodamine or alexa-647 (1:500; Thermo Fisher Scientific). Nuclei were counterstained with Hoechst 33258 (1:1000; Dojin, Tokyo, Japan). Images were captured using a Zeiss LSM 700 confocal microscope (Zeiss, Jena, Germany). The localization of certain frequencies in the cochlea used for whole-mount immunostaining was determined using the ImageJ (version 1.52a, NIH, New York, NY, USA) Measure Line plugin “https://www.masseyeandear.org/research/otolaryngology/eaton-peabody-laboratories/histology-core (accessed on 19 December 2019)”.

### 4.3. Morphological Analysis of Pillar Cells with Hematoxylin and Eosin Staining

Inner ears were fixed as above and decalcified for 7 days at 4 °C. Paraffin sections of 4 μm thickness were made in the horizontal plane parallel to the modiolus and stained with hematoxylin and eosin. Eight sections were analyzed for one mouse, and three mice were used for each genotype. The diameter of the stalk of outer pillar cells in the organ of Corti at the most basal and apical turns in the section was measured using ImageJ.

### 4.4. Auditory Brainstem Response and Distortion Product Otoacoustic Emission

Auditory brainstem response (ABR) and distortion product otoacoustic emission (DPOAE) measurements were performed as described previously [35]. Briefly, the mice were anesthetized by injecting a mixture of medetomidine (0.3 mg/kg), midazolam (4.0 mg/kg), and butorphanol (5.0 mg/kg). ABR waveforms were recorded at 5 dB sound pressure level (SPL) intervals at 8, 16, 24, and 32 kHz, decreasing from the maximum amplitude until the waveform, which is the average of 256 responses, could no longer be visualized. The threshold level was determined as the point above which any wave could be detected. Maximum sound pressure levels were set at 105, 107, 102, and 112 dB for 8, 16, 24, and 32 kHz, respectively. When no response was observed at the highest sound level available, the threshold was designated as 10 dB greater than that level.

DPOAEs were tested according to the primary tones, F1 and F2, which were set at an F2/F1 ratio of 1.22. The intensity of F1 (L1) was changed in 10 dB steps between 20 and 80 dB SPL, and the intensity of F2 (L2) was maintained at 10 dB under that of L1. The 2F1-F2 distortion components were estimated to constitute the DPOAE level. Hearing thresholds were estimated using L1-obtained 2F1-F2 as 0 dB. The probe for this measurement, which contained two receivers and one microphone, was developed and calibrated according to the EPL Acoustic System Assembly Manual (Massachusetts Eye and Ear Infirmary, Boston, MA, USA). A real-time processor (RP2.1, Tucker Davis Technology, Alachua, FL, USA) was used to generate the stimulus signals and digital–analog conversions of the recorded sound signals. Customized software based on LabVIEW version 2015 (National Instruments, Austin, TX, USA) was used to control the processor and for data analysis.

### 4.5. Cell Count

The areas of Rosenthal’s canal containing SGNs were outlined and measured using ImageJ. TUBB3-positive cells showing a nuclear Hoechst signal in Rosenthal’s canals were counted manually and expressed as the number per 10^4^ μm^2^.

### 4.6. Statistical Analysis

ABR data are presented as mean ± standard deviation (SD) and were analyzed with a two-tailed Student’s *t*-test using Microsoft Excel. Regarding the density of spiral ganglion neurons, two-way ANOVA and post hoc Tukey tests were performed using GraphPad Prism software version 9.5.0. Statistical significance was set at *p* < 0.05.

## Figures and Tables

**Figure 1 ijms-24-17092-f001:**
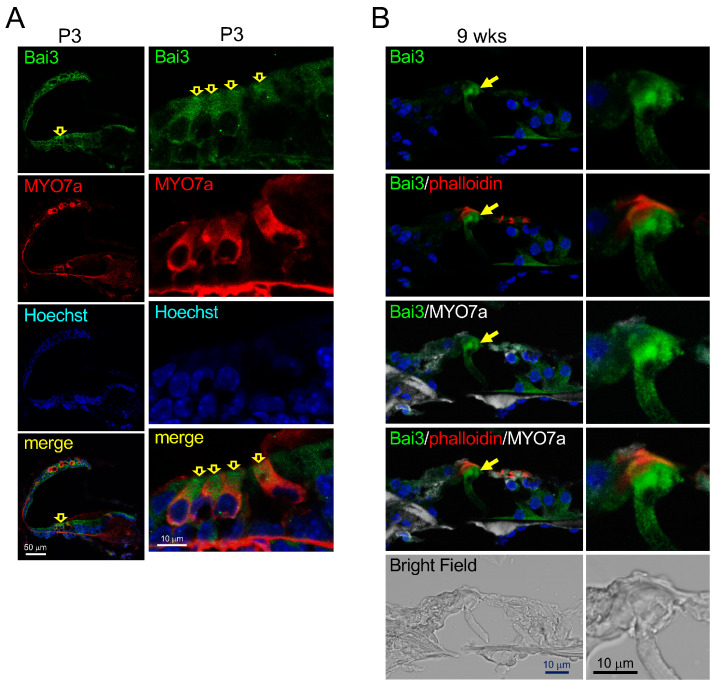
Expression of Bai3 in mouse cochleae. Immunosignals of Bai3 (green) were observed in hair cells of *Bai3*^+/+^ mice at P3 (**A**) (yellow hollow arrows), suggesting that Bai3 is expressed in cochlear hair cells of newborn mice. The left and right panels in (**A**) indicate the cochlea and organ of Corti, respectively. Tissues were co-immunostained with MYO7a (red). Nuclei were stained with Hoechst 33258 (blue). In the adult mouse cochlea, Bai3 immunosignals were detected in pillar cells (arrows in (**B**)). Tissues were co-immunostained with MYO7a (white) and phalloidin (red). Nuclei were stained with Hoechst 33258 (blue).

**Figure 2 ijms-24-17092-f002:**
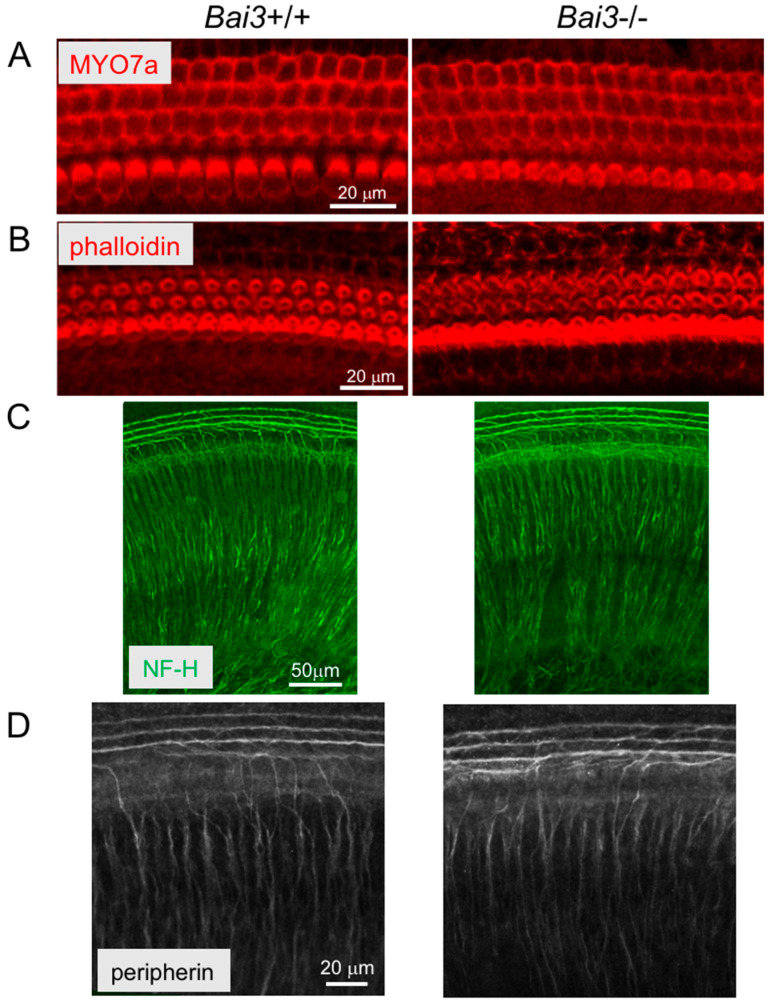
Normal development of hair cells and neuron innervation in *Bai3^-/-^* mice. (**A**,**B**) Whole-mount immunohistochemistry with the MYO7a antibody and phalloidin, respectively. The development of hair cells is not significantly different between *Bai3*^+/+^ and *Bai3^−/−^* mice at P3, and the innervation patterns of NF-H-positive (**C**) and peripherin-positive (**D**) neurons in the cochlea are not significantly different between *Bai3*^+/+^ and *Bai3^−/−^* mice at P3.

**Figure 3 ijms-24-17092-f003:**
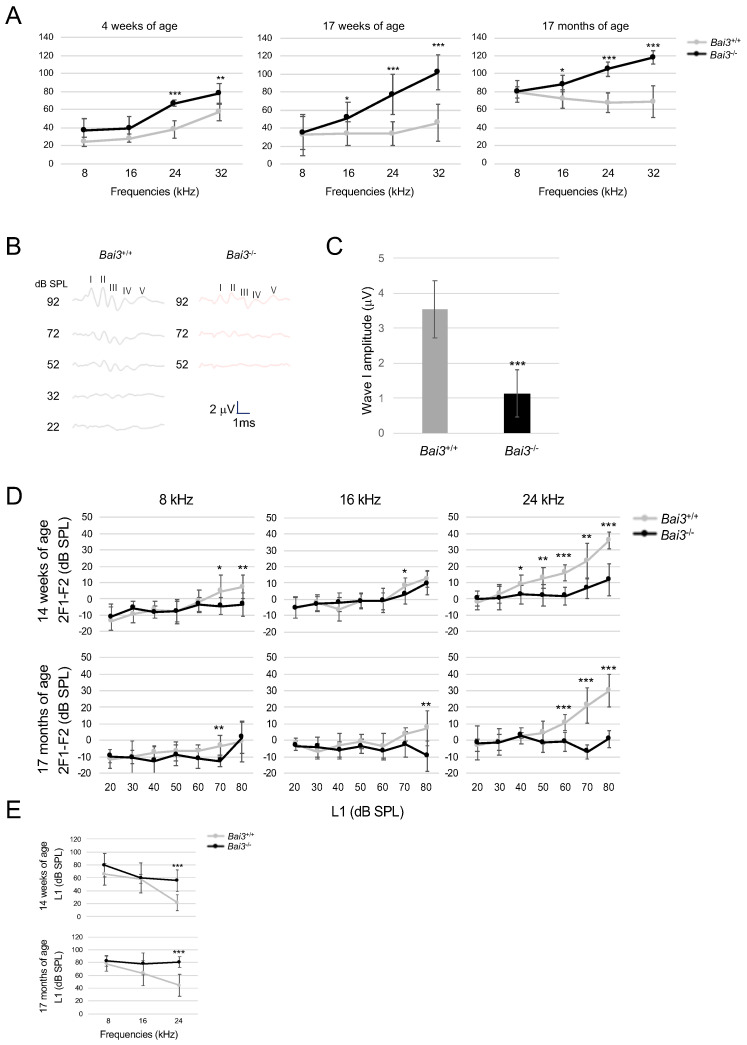
Thresholds of the auditory brainstem response and otoacoustic emission were increased in a frequency-dependent manner in *Bai3^-/-^* mice. (**A**) To study the function of Bai3 in the auditory system, we measured auditory brainstem responses (ABRs), which represent the sound-evoked neural output of the cochlea. Thresholds are elevated at 16 kHz and higher frequencies in *Bai3^−/−^* mice compared to *Bai3*^+/+^ mice. *Bai3*^+/+^ (4 weeks of age), n = 8 ears from four mice; *Bai3^−/−^* (4 weeks), n = 6 ears from three mice; *Bai3*^+/+^ (17 weeks), n = 8 ears from four mice; *Bai3^−/−^* (17 weeks), n = 12 ears from six mice; *Bai3*^+/+^ (17 months), n = 6 ears from three mice; *Bai3^−/−^* (17 months), n = 8 ears from four mice. Error bars indicate the standard deviation. *, *p* < 0.05; **, *p* < 0.01; ***, *p* < 0.001. Statistical significance was analyzed using a t-test between *Bai3*^+/+^ and *Bai3^−/−^* mice at each frequency. (**B**) Representative ABR measurements at 24 kHz in *Bai3*^+/+^ (left) and *Bai3^−/−^* (right) mice at 14 weeks of age. Roman numerals mark the peaks of the standard ABR waves. Averages of wave I amplitude (maximum sound pressure level, 24 kHz) are shown in (**C**). *Bai3*^+/+^ (17 weeks), n = 8 ears from four mice; *Bai3^−/−^* (17 weeks), n = 12 ears from six mice. Error bars indicate the standard deviation. *** *p* < 0.001. Our data show the lower amplitude of wave I in *Bai3^−/−^* vs. *Bai3*^+/+^, indicating that the elevated threshold of ABR is, at least in part, due to cochlear damage. Delayed latency of later waves was not observed in *Bai3^−/−^* mice. (**D**) Distortion product otoacoustic emission (DPOAE) measurements were performed at 8, 16, and 24 kHz. DPOAE thresholds were determined as the L1 dB SPL when the 2f1-f2 was 0 dB, and average DPOAE thresholds are shown in (**E**). *Bai3*^+/+^ (14 weeks), n = 8 ears from four mice; *Bai3^−/−^* (14 weeks), n = 14 ears from seven mice; *Bai3*^+/+^ (17 months), n = 10 ears from five mice; *Bai3^−/−^* (17 months), n = 8 ears from four mice. Statistical significance was analyzed using a t-test between *Bai3*^+/+^ and *Bai3^−/−^* at each frequency. Error bars indicate the standard deviation. *, *p* < 0.05; **, *p* < 0.01; *** *p* < 0.001.

**Figure 4 ijms-24-17092-f004:**
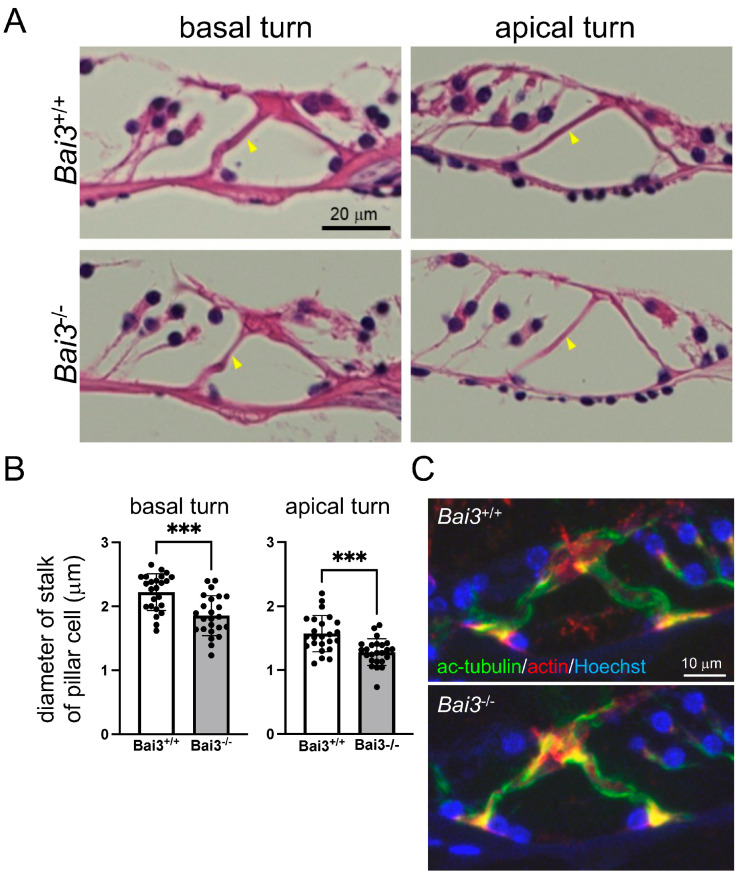
Thinner pillar cells in *Bai3^−/−^* adult mice. (**A**) HE staining of cochlea showed that stalks of outer pillar cells were thinner in *Bai3^−/−^* mice than in *Bai3*^+/+^ mice (arrowheads). (**B**) Quantitation of diameter of stalks of outer pillar cells. Error bars indicate the standard deviation. n = 24 sections from three mice in each group. *** *p* < 0.001. (**C**) Sections of organ of Corti at basal cochlear turn were immunostained with acetylated tubulin (green) and beta-actin (red) antibodies. We did not detect any significant differences in the expression pattern of either acetylated tubulin or beta-actin between *Bai3*^+/+^ and *Bai3^−/−^* mice at 11 weeks of age. Nuclei were stained with Hoechst 33258 (blue).

**Figure 5 ijms-24-17092-f005:**
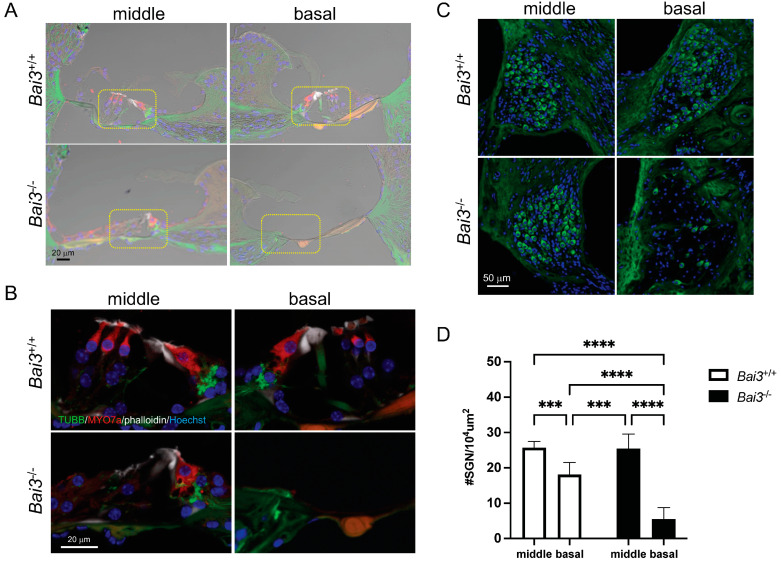
Degeneration of hair cells and spiral ganglion neurons in old *Bai3^−/−^* mice. (**A**) Cochlear sections were immunostained with neuron-specific class III b-tubulin (TUBB3) antibody (green), MYO7a antibody (red), and phalloidin (white). Fluorescent images were overlaid with bright-field images. The organ of Corti (yellow dotted line) is missing at the basal turn of the *Bai3^−/−^* cochlea (bottom right). Nuclei were labeled with Hoechst 33258 (blue). Scale bar: 20 μm. Regions marked by the yellow dotted line in (**A**) are enlarged in (**B**) without the bright-field image. MYO7a-positive HCs (red) are lost at the basal turn of *Bai3^−/−^* cochleae. (**C**) Cochlear sections were immunostained with TUBB3 antibody (green). Scale bar: 50 μm. Loss of TUBB-positive SGNs is observed in the basal turn of *Bai3^−/−^* cochleae (bottom right). (**D**) Quantitation of SGN density at the basal and middle turns of the cochlea of mice at 14–17 months of age. The number of TUBB-positive cells in Rosenthal’s canal was counted and expressed as #SGNs per 10^4^ μm^2^. Data are presented as mean ± standard deviation. n = 9 sections from three mice in each group. ***, *p* < 0.001; ****, *p* < 0.0001.

## Data Availability

The data supporting the findings of this study are openly available within the article or from the corresponding author upon request.

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
