# Peer review of "Brain-Specific Angiogenesis Inhibitor 3 Is Expressed in the Cochlea and Is Necessary for Hearing Function in Mice"

_ijms, 2023, doi:10.3390/ijms242317092_

Round 1
Reviewer 1 Report (Previous Reviewer 1)
Comments and Suggestions for Authors
Revised Review Reports:
The authors have responded to my concern satisfactory and I think the revised paper now is suitable for publication.

Author Response
Thank you for your comments.
Reviewer 2 Report (Previous Reviewer 2)
Comments and Suggestions for Authors
The manuscript entitled “Brain-specific angiogenesis inhibitor 3 is expressed in the cochlea and is necessary for hearing function in mice” aims to investigate the expression and function of brain-specific angiogenesis inhibitor 3 (Bai3) in the cochlea and its role in the development and maintenance of the tunnel of Corti, a crucial structure for proper hearing function in mice. The manuscript is very interesting and fits well with the scope of the journal. It is well-prepared and also improved compared to the previous version. I still have some suggestions for improvement since not all my comments were addressed.
While the introduction provides background information on hearing loss and the importance of understanding the inner ear structure, it could be more explicit in stating the specific research gap or the need for investigating the role of Bai3 in the auditory system. Clearly articulate why studying Bai3 in the context of the cochlea and hearing function is important and what knowledge gaps this study aims to address.
Also, it would be helpful to expand on the relevance of Bai3 in other systems and tissues to set the stage for its potential role in the auditory system.
While the introduction indicates that the study aims to analyze Bai3 expression in the cochlea and study the hearing function of Bai3 knockout mice, it would be beneficial to explicitly state the specific research objective in a clear and concise manner. The motivation for the work and novelty should be clearly stated at the end of the Introduction.
Figure 3 should be modified since it is crowded.
It would be helpful to discuss any limitations of the study and potential areas for future research. For instance, if there were challenges in identifying specific cellular or molecular changes related to Bai3 deficiency, acknowledge these limitations and propose potential alternative approaches or techniques that might be useful for further investigations.
Comments on the Quality of English LanguageMinor changes required.
Author Response
Please see the attachment.

This manuscript is a resubmission of an earlier submission. The following is a list of the peer review reports and author responses from that submission.
Round 1
Reviewer 1 Report
Comments and Suggestions for Authors
This manuscript demonstrated that that Bai3 deficiency causes abnormal formation of the tunnel of Corti and elevated hearing thresholds at high frequencies. In addition, they have shown degeneration of HC and SGN in the basal rotation of the cochlea of old Bai3-/- mice. These new findings suggest that cochlear Bai3 is required for normal auditory function.
In this paper, the authors speculate that Bai3 deficiency, which has not been studied before, causes congenital and progressive hearing loss, that hearing loss occurs predominantly at high frequencies, and that the cause is due to abnormalities in the tunnel of Corti morphology, based on histological studies. The reviewer believes that the study is worth reporting because it reveals a new inner ear pathology, especially the frequency specificity of the hearing impairment despite the fact that it is thought to be caused by abnormalities in the morphology of the tunnel of Corti. However, the reviewer feels that the manuscript unfortunately does not contain enough experimental results to support the authors' speculation. Therefore, the reviewer would like to ask the authors to make a effort to add significant additions to the results for the concerns pointed out below. In some cases, the authors may need additional experiments in order to respond to the reviewer's request. If additional experiments are not feasible in terms of animal availability, etc., we would like to see the authors’ possible response by making maximum use of existing images, etc.
Major comments;.
First of all, the authors did not indicate the strain of the mouse used in the experiment. However, this information is extremely important, especially for auditory research on mice, and the strain must be clearly indicated in the manuscript, and the problems that the C57/BL6 has when auditory research been conducted using this strain must be described.
This manuscript discusses progressive hearing loss with a predominance of high frequencies, but Figures 1 and 2 did not show any frequency information in the cochlea of both Bai3+/+, and Bai3-/-. For example, Figure 1D was presumed to be the morphology of the organ of Corti in the region close to the basal turn, but it seems impossible to discuss the subsequent frequency specificity without observing and comparing Bai3 expression in the apex side organ or Corti. In the case of the cross section, the authors must have images other than the basal turn, and Figure 1 and Figure 2 must be reconstructed by adding these picures.
In Figure 3, ABR and DPOAE were measured in young and old Bai+/+ and Bai-/-. However, C57/BL6 at 17 weeks old is not considered as a young age for auditory studies, and the strain's specific auditory deficits at high frequencies should be already present. Therefore, it is not possible to determine whether the hearing impairment in the high frequencies observed at 17 weeks old was congenital or whether it was an acquired elevation of the hearing threshold that was more likely to occur than in the wild type. If possible, ABRs and DPOAEs from 3 to 5 weeks old should be presented to accurately estimate whether the hearing impairment was congenital or acquired.
The reviewer felt that the DPOAE data shown in Figure 3 was extremely problematic. In this study, DPOAE at 16 kHz as well as 8 kHz was hardly induced in 14 week old Bai+/+. This would seem to indicate that the DPOAE measurement system used in this study did not have the necessary accuracy to be used in mouse studies. It seems not to be possible to assess outer hair cell function. An appropriate response is needed, either replacing with appropriate data or removing the DPOAE results from this manuscript.
P6, L210 stated "the tunnel of Corti was not properly formed in the cochlea of some Bai3-/- mice compared to those of Bai3+/+ mice". However, only "some of mice" in which changes were extremely observed were shown in the Figure. Thus, showing only the results expected by the authors is not a sincere attitude toward the research. It is necessary to specify what proportion of the individuals had collapsed tunnel of Corti, and whether the uncollapsed tunnel of Corti was similar to Bai3+/+ or not.
In Figure 4A, the Tunnel of Corti appeared to collapse more in the pictures shown in Bai3+/+. If there is a mistake in such an important finding, it is necessary to review the entire figure because it will affect the reliability of the entire manuscript.
Figure 4B showed a representative example of a collapsed tunnel of Corti, which to the reviewer appears to be an artifact during OCT or sectioning. The morphological findings of Bai3-/- is a very important and should be shown to be reproducible.
Minor comments;.
The ABR threshold in Figure 3 was described in the text as "no measurable response at the highest stimulus levels tested," but it did not state what the sound pressure was and how the results beyond the maximum sound pressure level were handled. This should be clearly stated in the Materials and methods section.
The low quality of the surface preparation image in Figure 4C makes it difficult to determine whether or not there was an abnormality in this shown organ of Corti. It would be better if the reviewer could replace the panel with one of higher quality.
Reviewer 2 Report
Comments and Suggestions for Authors
The manuscript entitled “Brain-specific angiogenesis inhibitor 3 is expressed in the cochlea and is necessary for hearing function in mice” aims to investigate the expression and function of brain-specific angiogenesis inhibitor 3 (Bai3) in the cochlea and its role in the development and maintenance of the tunnel of Corti, a crucial structure for proper hearing function in mice. The manuscript is very interesting and fits well with the scope of the journal. It is well-prepared in general, but I have some suggestions for improvement.
The abstract could benefit from briefly mentioning the significance of understanding the molecular mechanisms underlying the development and maintenance of the tunnel of Corti. Why is this important in the broader context of hearing research or potential implications for human hearing disorders?
Besides, while the abstract hints at the aim of the study, it could be more explicit about the main research question or objective. For example, explicitly state that the study aimed to investigate the expression, function, and role of Bai3 in the cochlea and its impact on the tunnel of Corti and hearing function in mice.
Also, the abstract provides a general overview of the results, but it could be strengthened by including some specific key findings.
While the introduction provides background information on hearing loss and the importance of understanding the inner ear structure, it could be more explicit in stating the specific research gap or the need for investigating the role of Bai3 in the auditory system. Clearly articulate why studying Bai3 in the context of the cochlea and hearing function is important and what knowledge gaps this study aims to address.
Also, it would be helpful to expand on the relevance of Bai3 in other systems and tissues to set the stage for its potential role in the auditory system.
While the introduction indicates that the study aims to analyze Bai3 expression in the cochlea and study the hearing function of Bai3 knockout mice, it would be beneficial to explicitly state the specific research objective in a clear and concise manner.
The results are well-presented in general, but Figures 1, 4, and 5 are a bit overwhelming. Maybe they could be partially transferred to the supplement?
The discussion effectively summarizes the key findings, including the abnormal formation of the tunnel of Corti, elevated hearing thresholds at high frequencies, and the degeneration of hair cells and spiral ganglion neurons in old Bai3-/- mice. Emphasizing these main findings early in the discussion could be useful.
While the discussion describes the results in detail, it could further elaborate on the implications of these findings. Specifically, explain how the observed abnormalities in the tunnel of Corti and the degeneration of key auditory components (hair cells and spiral ganglion neurons) may directly contribute to the hearing impairment observed in Bai3-/- mice. Additionally, discuss how these findings align with or add to existing knowledge in the field.
The discussion mentions that young adult Bai3-/- mice showed elevated hearing thresholds at high frequencies despite the absence of histological abnormalities in hair cells and spiral ganglion neurons. It would be valuable to offer possible explanations for this discrepancy.
It would be helpful to discuss any limitations of the study and potential areas for future research. For instance, if there were challenges in identifying specific cellular or molecular changes related to Bai3 deficiency, acknowledge these limitations and propose potential alternative approaches or techniques that might be useful for further investigations.
Also, the discussion could elaborate on the potential clinical relevance of the findings in Bai3-/- mice for human hearing disorders. Discuss how the insights gained from this animal model might inform research on human sensorineural hearing loss and whether Bai3 or related pathways could serve as therapeutic targets for hearing restoration.
Comments on the Quality of English LanguageMinor editing is required.